# Multifaceted Antibiotic Resistance in Diabetic Foot Infections: A Systematic Review

**DOI:** 10.3390/microorganisms13102311

**Published:** 2025-10-06

**Authors:** Weiqi Li, Oren Sadeh, Jina Chakraborty, Emily Yang, Paramita Basu, Priyank Kumar

**Affiliations:** 1Touro College of Pharmacy, New York, NY 10036, USA; wli11@student.touro.edu; 2New York College of Podiatric Medicine, Touro University, New York, NY 10035, USA; osadeh2025@nycpm.edu; 3Oxford College, Emory University, Oxford, GA 30054, USA; jinachakraborty@gmail.com; 4Department of Foundational Sciences, College of Veterinary Medicine, Western University of Health Science, Pomona, CA 91766, USA; emily.yang@westernu.edu

**Keywords:** diabetic foot infection, antibiotic resistance, multidrug resistance, comorbidities, bacterial pathogens, antimicrobial stewardship, Gram-negative bacteria, diabetic foot ulcer

## Abstract

Diabetic foot infections (DFIs) are a significant complication in patients with diabetes, often leading to severe clinical complications including amputation and increased mortality rates. The effective management of these infections is complicated by the rise in antibiotic resistance among the microbial populations involved. In this paper, we undertake a systematic review and meta-analysis to explore the bacterial profiles, as well as their antibiotic resistance patterns in DFIs, encompassing studies published between 2014 and 2024. A total of 28 studies were selected from several databases, including PubMed, Google Scholar, EBSCOhost, and ScienceDirect, published from 2014 to 2024, specifically focusing on diabetic foot infections and antibiotic resistance. Diabetic foot infections arise from a combination of factors, including peripheral neuropathy, poor circulation, and immune system impairment, making diabetic patients prone to unnoticed injuries, impaired wound healing, and a higher risk of infections. The severity of DFIs often depends on the size and depth of the ulcers, with larger, deeper ulcers posing additional risks of infection and complications, such as osteomyelitis and sepsis. Our study synthesizes information on the total isolates of microbes, their resistance to one or more groups of antibiotics, and resistance panel results across multiple antibiotics, including amoxicillin/clavulanate, trimethoprim/sulfamethoxazole, ciprofloxacin, and others. We meticulously catalog the resistance of key bacterial strains—*Escherichia coli*, *Enterobacter* spp., *Proteus* spp., *Pseudomonas* spp., *Staphylococcus aureus*, and others—highlighting patterns of resistance to single and multiple antibiotic groups. This systematic review also analyzes the correlations of various comorbidities reported by the diabetic foot infection patient populations in the included studies with multiple antibiotic resistance patterns. Subsequently, this analytical review study addresses the rising prevalence of antibiotic-resistant pathogens and underscores the need for antibiotic stewardship programs to promote judicious use of antibiotics, reduce the spread of resistant strains, and enhance therapeutic outcomes. In addition, the review discusses the implications of resistance to empirical antibiotic treatments, underscoring the necessity for tailored antibiotic therapy based on culture and sensitivity results to optimize treatment outcomes.

## 1. Introduction

Diabetic foot infections (DFIs) represent a significant and severe complication for individuals with diabetes mellitus, leading to substantial morbidity and increased healthcare costs worldwide [1]. These infections arise due to a combination of factors, including neuropathy, poor circulation, and immune system impairment, which collectively facilitate the development and progression of wounds that become infected [2,3]. Diabetes mellitus, particularly Type 2 diabetes, is most commonly associated with DFIs, though Type 1 diabetes can also lead to these complications [4]. The pathophysiology of diabetes, characterized by chronic hyperglycemia, underpins the development of DFIs. Peripheral neuropathy in diabetic patients leads to a loss of sensation, making them prone to unnoticed injuries, while peripheral arterial disease (PAD) and impaired immune function hinder wound healing and increase the risk of infection [5]. The location of DFIs is typically on weight-bearing areas of the foot, such as the plantar surface, the toes, and areas prone to repetitive trauma, such as bony prominences or deformities [6]. These high-pressure areas are especially vulnerable to developing ulcers that can progress to infection. Additionally, the size and depth of diabetic foot ulcers are critical risk factors for infection; larger, deeper ulcers are more likely to become infected and lead to complications such as osteomyelitis or sepsis [6]. Foot deformities and poor circulation further contribute to the high risk of infection. These factors, when combined, create an environment in which wounds heal slowly, allowing infections to progress unchecked.

Once established, DFIs often result in prolonged hospital stays, frequent surgical interventions, and in severe cases, amputations, dramatically affecting patients’ quality of life and imposing a heavy burden on healthcare systems globally [1]. Early diagnosis and timely intervention are essential to improving patient outcomes. Multidisciplinary care, including podiatric, vascular, and infectious disease specialists, can significantly reduce the risk of severe complications. Furthermore, patients with DFIs frequently have comorbidities such as cardiovascular disease, renal impairment, and PAD, which complicate the management and treatment outcomes.

A critical challenge in managing DFIs is the rising prevalence of antibiotic resistance among the pathogens responsible for these infections. The misuse and overuse of antibiotics have accelerated the emergence of resistant strains, complicating treatment regimens and reducing the efficacy of standard therapeutic options [7]. Antibiotic stewardship programs play a pivotal role in addressing this issue by promoting the judicious use of antibiotics, optimizing therapeutic strategies, and minimizing the development of resistance. These programs involve implementing evidence-based guidelines for the selection, dosing, and duration of antibiotic therapy, with the goal of improving patient outcomes while preserving the efficacy of existing antibiotics [8].

Understanding the patterns of antibiotic resistance in DFIs is crucial for optimizing antibiotic therapy, improving patient outcomes, and informing guidelines for clinical practice. This analysis aims to provide a comprehensive overview of the current landscape of antibiotic resistance in DFIs to explore how geographic factors influence the bacterial profiles and resistance patterns in DFIs across different regions, identify trends and gaps in the literature, and offer evidence-based recommendations for clinicians and policymakers in the management of these challenging infections. Furthermore, it underscores the need to incorporate antibiotic stewardship into routine clinical care to curb resistance and enhance treatment effectiveness.

## 2. Methods

### 2.1. Study Design

This study was designed as a systematic review and limited meta-analysis to synthesize available data on bacterial profiles and antibiotic resistance in diabetic foot infections (DFIs). Descriptive statistical analysis was conducted to combine data where possible, providing a comprehensive estimate of antibiotic resistance patterns in DFIs from 2014 to 2024.

### 2.2. Search Strategy

A comprehensive search was conducted in four major electronic databases—PubMed, Google Scholar, EBSCOhost, and ScienceDirect—to identify relevant studies. The search terms used included combinations of the following keywords: “diabetic foot infections,” “antibiotic resistance,” “DFI microbial isolates,” “diabetes-related foot infections,” “bacterial resistance,” and “diabetic foot ulcer infections.”

The initial search yielded 1900 results from Google Scholar, 63 results from EBSCOhost, 20 results from PubMed, and 37 results from ScienceDirect. After duplicates were removed, the remaining titles and abstracts were screened for relevance. Full-text articles were then reviewed based on predefined inclusion and exclusion criteria (detailed below). The final number of included studies was 28. The search strategy used to guide the selection process of the studies included in the systematic review is indicated in the PRISMA diagram shown in Figure 1 [9]. The abbreviated version of the PRISMA checklist was followed (provided as a Appendix A) [10].

### 2.3. Inclusion and Exclusion Criteria

The following criteria were used to determine the eligibility of studies:

Inclusion Criteria:Studies published between January 2014 and June 2024.Studies that provided data on the microbiological profile of diabetic foot infections and reported antibiotic resistance patterns.Research conducted on human subjects diagnosed with DFIs, either in hospital or outpatient settings.Full-text studies that analyzed bacterial isolates from DFIs, particularly focusing on resistance to antibiotics such as amoxicillin/clavulanate, trimethoprim/sulfamethoxazole, ciprofloxacin, and other commonly used antibiotics.

Exclusion Criteria:Review or meta-analysis papers.Case reports, commentaries, and opinion pieces.Studies without available full-text or studies not reporting data relevant to antibiotic resistance in DFIs.Articles that did not provide conclusive data on DFI-related microbial resistance or clinical outcomes.

### 2.4. Data Extraction and Analysis

Two independent reviewers (WL, EW) extracted data from the included studies using a standardized data collection form. Discrepancies were resolved through discussion or by consulting a third reviewer (OS). The following data were extracted from each study:Study characteristics: author(s), year of publication, country, study design, and sample size.Patient characteristics: age, sex, type of diabetes (Type 1 or Type 2), duration of diabetes, and comorbidities such as peripheral neuropathy, peripheral arterial disease, and renal impairment.Clinical factors: type of infection (e.g., diabetic foot ulcer, osteomyelitis), severity of the infection (e.g., depth, size of ulcer), and history of previous infections or treatments.Microbiological data: prevalence and type of bacterial isolates, including *Staphylococcus aureus* (MRSA and MSSA), *Pseudomonas* spp., *Proteus* spp., *Escherichia coli*, *Enterobacter* spp., and others.Antibiotic resistance data: rates of resistance to specific antibiotics such as amoxicillin/clavulanate, ciprofloxacin, vancomycin, carbapenems, cephalosporins, and aminoglycosides.

Analyses were performed by three investigators (WL, PK, PB), two of whom were assigned randomly to each study to reduce bias.

### 2.5. Quality Assessment and Risk of Bias Analysis

Risk of bias was reduced in the methodology by independent review of the included studies and data extraction and analysis. The quality of the included studies was assessed using the PRISMA (Preferred Reporting Items for Systematic Reviews and Meta-Analyses) guidelines [10] using a modified version of the Cochrane ROBINS-E tool [11] that was customized for this analysis [12].

The methodological quality of the studies was evaluated based on the following criteria:The clarity of research questions and objectives.The appropriateness of study designs and sample sizes.The robustness of microbiological testing methods for identifying bacterial isolates and antibiotic resistance.The reporting of relevant clinical outcomes, including treatment success and recurrence rates.The risk of bias in individual studies was assessed using a modified version of the Cochrane risk of bias tool for non-randomized studies. This included 20 components which were specifically developed for this study under the subheadings: research question, selection criteria, participant characteristics, sample size, outcome, methods, and analysis of which is more relevant for DFIs and antibiotic resistance research conducted as observational prospective studies. The questions were first trialed on the excluded articles to assess the tool validity. Two investigators (P.B. and J.C.) conducted the quality assessment and any discrepancies were resolved by discussion. All 20 questions were equally weighted, and individual scores were calculated based on the proportion of “yes” answers. Studies that scored <50%, 50–75%, and >75% were deemed of high, moderate, and low risk of bias, respectively, based on a collective decision taken by the investigators in line with the ROBINS-E tool guidelines [11,12].

### 2.6. Statistical Analysis

Where possible, a systematic review and meta-analysis was performed to pool the results from individual studies. We used R software (version 4.4.1) for statistical analysis. The systematic review focused on combining the prevalence of key bacterial isolates and their antibiotic resistance rates. Depending on the heterogeneity of the studies, either a fixed-effect model or a random-effects model was employed.

Heterogeneity Assessment: The I^2^ statistic was calculated to assess the degree of heterogeneity across studies included in this systematic review. An I^2^ value greater than 50% indicated substantial heterogeneity, in which case a random-effects model was used. Forest plots were generated to visually represent the pooled estimates of resistance for individual bacterial species and their resistance to specific antibiotics. We used Julius AI to generate heat maps showing extent and significance of correlation.

### 2.7. Outcomes of Interest

The primary outcome of interest was the prevalence of antibiotic-resistant bacteria in DFIs. Secondary outcomes included:The impact of comorbidities (e.g., vascular disease, renal impairment) on antibiotic resistance patterns. Correlation analysis was performed using R.The relationship between prior antibiotic use, hospitalization, and the development of resistant infections.The identification of risk factors for multidrug-resistant organisms (MDROs) in DFIs.

## 3. Results

Table 1 provides a summary list of the 28 references that reported the pathogens and their resistance related to diabetic infection and their source database.

### 3.1. Analysis of Study Populations

The demographic breakdown of study participants presents a gender distribution of the populations in the included studies. 63% of participants were male, indicating a male-dominant study sample while 37% were female, suggesting a possible underrepresentation of women in the data (Figure 2A).

The geographic distribution of study populations reveals considerable variation in study participant numbers across 22 countries. China contributed the largest share, with nearly 1000 participants, followed closely by Malaysia with approximately 750. Turkey and Peru also had substantial contributions, with around 400 and 300 participants, respectively. A moderate level of participation (100–250) was seen in countries such as Italy, Lebanon, South India, Brazil, and India. Lower study populations (fewer than 100) were recorded in Oman, Greece, Vietnam, Columbia, Iran, and Pakistan, indicating more limited study engagement or data availability in these regions (Figure 2B) [34,35].

### 3.2. Epidemiological Analysis of Pathogens Related to Comorbidity

The prevalent comorbidities reported in the study populations with DFIs showing resistance to one or more antibiotics in each of the 28 studies included in this analysis were compiled and compared (Figure 3). Among the 5096 patients included in the study, the most prevalent comorbidity was neuropathy, affecting 18.4% of the cohort (*n* = 939). Hypertension (HTN) was the second most common, present in 16.5% of patients (*n* = 839), followed closely by peripheral vascular disease, which was observed in 16.0% of the population (*n* = 816). Notably, nephropathy affected 10.1% of patients (*n* = 514), further complicating the management of DFIs. Additionally, 7.3% of the cohort (*n* = 370) had a history of previous ulceration, while 6.5% (*n* = 333) had undergone previous amputations.

### 3.3. Epidemiological Analysis of Pathogens Related to Antibiotic Resistance

On compilation and comparing the prevailing microbial pathogen species isolated from the study populations with DFI showing resistance to one or more antibiotics (Figure 4) [34,35,36,37], *Staphylococcus aureus* was identified as the most prevalent bacterial isolate, with a total of 1336 isolates, followed by *Pseudomonas* spp. (955 isolates), *Enterococcus* spp. (533 isolates), *Enterobacter* spp. (693 isolates), and *Escherichia coli* (674 isolates). These pathogens represent the top bacterial isolates in diabetic foot infections.

When analyzing the number of microbial isolates demonstrating resistance to a single class of antibiotics, several important trends were observed: *Staphylococcus aureus* showed the highest frequency of single antibiotic resistance, with 64 isolates resistant to at least one antibiotic group (Figure 5A), suggesting that *S. aureus* is a major contributor to mono-resistant infections. MRSA (Methicillin-Resistant *Staphylococcus aureus*) and MSSA (Methicillin-Sensitive *S. aureus*) are separately listed, with moderate numbers of resistance. *Pseudomonas* spp. showed the second-highest resistance burden, with 16 isolates resistant to one antibiotic group, reflecting its well-known intrinsic and acquired resistance mechanisms. Other Gram-negative organisms (e.g., *Escherichia coli*, *Klebsiella* spp., *Proteus* spp., *Enterobacter* spp.) show lower but significant resistance as well, while Coagulase-negative staphylococci (CoNS) and *Streptococcus* spp. also showed notable resistance. These findings indicate that these pathogens may be early indicators of broader resistance development and are increasingly difficult to treat with standard antibiotics.

Regarding resistance to two or more antibiotic groups, which analyze the burden of multidrug resistance (MDR) across a range of microbial species, *Acinetobacter* was the most resistant, with 274 isolates showing multidrug resistance (Figure 5B). This confirms *Acinetobacter*’s reputation as a “superbug” in healthcare settings. All the other major Gram-positive DFI pathogens, *Staphylococcus aureus*, CoNS, *Streptococcus* spp., and *Enterococcus* spp., all showed MDR levels between 150–200 MDR isolates with 170 resistant isolates of *S. aureus*. *Escherichia coli* (107 multidrug-resistant isolates) and *Morganella morganii* showed moderate MDR levels (80–120 range), while *Pseudomonas* spp. also displayed moderate levels of multidrug resistance, with 60 isolates resistant to two or more antibiotic groups, despite its intrinsic resistance. *Proteus*, *Klebsiella*, *Enterobacter* spp. displayed lower but still concerning levels of MDR.

When combining the insights obtained from analyzing antibiotic resistance, several important trends were observed after comparing the frequency of prevalence of resistance to one antibiotic group and that of resistance to 2 or more antibiotic groups. This comparison highlights emerging resistance patterns and the clinical risks posed by both mono-resistant and multidrug-resistant (MDR) organisms across various microbial species (Table 2). *Staphylococcus aureus* and CoNS show notable transition from monoresistance to MDR, while *Pseudomonas* and *E. coli* also demonstrate moderate dual risk. *Corynebacterium* spp. and *Enterococcus* spp. show low monoresistance but high MDR, indicating resistance develops rapidly or is under detected in the early stage. *Acinetobacter* spp. has low monoresistance but highest MDR, underscoring its role as a nosocomial superbug.

### 3.4. Correlation Between Types of Comorbidities Found in DFI and Types of Antibiotic Resistance

On analysis of interdependence between pre-existing comorbidities and antibiotic resistance prevalence in DFI, strong positive correlations were observed between the *Staphylococcus* strain’s resistance to penicillin and the prevalence of nephropathy (r = 0.956) and hypertension (r = 0.905). Dyslipidemia shows the strongest correlations with multiple antibiotic resistance patterns, with very strong positive correlation with clindamycin resistance (r ≈ 0.997), gentamicin resistance (r ≈ 0.986), erythromycin resistance (r ≈ 0.973), penicillin resistance (r ≈ 0.949) and with vancomycin resistance (r ≈ 0.978) (Figure 6). These findings suggest that as comorbidity prevalence increases, so does antibiotic resistance (Table 3).

All these correlations are statistically significant (*p* < 0.05). The heatmap visualizes the full correlation matrix, with red indicating positive correlations and blue indicating negative correlations. The intensity of the color represents the strength of the correlation. Similar correlation analysis performed between the comorbidities with resistance patterns to all the groups of antibiotics that were reported in the studies included in this systematic review for *Pseudomonas*, *Enterococcus*, *E. coli*, and *Proteus* also showed varying degrees of correlations. Only the pairs that yielded correlation coefficients of up to moderate degree of significance are included in Table 4. The summarized data presented shows that smoking (current or former) is the strongest predictor of resistance across multiple pathogens, including *Proteus*, *E. coli*, *Pseudomonas*, and *Enterococcus*. Dyslipidemia drives extensive resistance in *S. aureus*. Nephropathy, PAD, hypertension, and prior amputations are major risk factors for MDR, especially in Gram-negative pathogens. *Proteus* spp. shows the highest correlation strength (r > 0.99) with multiple comorbidities, suggesting high-risk resistance profiles in these infections.

Detailed correlation analysis and visual interpretation of the correlation of all the comorbidities with resistance patterns to all the groups of antibiotics that were reported in the studies included in the systematic review are shown in the Appendix A.

This study analyzed the correlation between patient comorbidities and antibiotic resistance patterns in bacterial pathogens isolated from diabetic foot infections (DFIs). Across all pathogens, several comorbidities—particularly dyslipidemia, hypertension, nephropathy, peripheral artery disease (PAD), smoking, and history of amputation—showed statistically significant associations with increased antimicrobial resistance (*p* < 0.01).

*Staphylococcus aureus* exhibited very strong correlations with dyslipidemia, which was linked to resistance against clindamycin, gentamicin, vancomycin, erythromycin, and penicillin (r ≈ 0.949–0.997). Hypertension was also significantly associated with resistance to erythromycin (r ≈ 0.883), indicating that metabolic and cardiovascular conditions may facilitate resistance in *S. aureus*.

*Pseudomonas* spp. showed pronounced resistance in patients with neuropathy, which strongly correlated with resistance to multiple drug classes including cephalosporins, carbapenems, and quinolones. Current smoking, PAD, nephropathy, and hypertension also showed high correlations with resistance to piperacillin-tazobactam, imipenem, erythromycin, and gentamicin. These results highlight the substantial influence of neurovascular complications and lifestyle factors in driving resistance in Pseudomonas isolates.

*Enterococcus* spp. demonstrated strong positive correlations between resistance (including vancomycin resistance) and comorbidities such as smoking and hypertension. Nephropathy and PAD contributed to moderate associations with resistance to tetracycline and gentamicin, implicating vascular and renal disease in the persistence of vancomycin-resistant *Enterococcus* (VRE).

*Escherichia coli (E. coli)* resistance was most strongly linked to current smoking status, which correlated with resistance to six antibiotic classes, including cephalosporins and aminoglycosides. Additionally, nephropathy, previous amputation, and hypertension were moderately associated with multidrug resistance, especially to gentamicin, carbapenems, and tetracycline.

*Proteus* spp. showed the highest correlation strength overall. Current smoker status had near-perfect correlations (r > 0.99) with resistance to erythromycin, piperacillin-tazobactam, imipenem, and cephalosporins. Previous amputation (r ≈ 0.99) and nephropathy (r ≈ 0.98) were also strongly associated with resistance to gentamicin and aminoglycosides, respectively. Former smokers exhibited a strong association with tetracycline resistance (r ≈ 0.99).

## 4. Discussion

This study highlights several important findings regarding the bacterial profiles and comorbidities associated with DFIs. The geographic and demographic distribution of the populations in the studies published in the last 10 years reporting antibiotic resistance in DFIs included in this systematic review show that there is a significant geographical disparity in study population sizes, with certain countries like China and Malaysia contributing disproportionately larger sample sizes. These differences could reflect factors such as available research funding, population size, healthcare infrastructure, or prioritization of the condition studied in different regions.

*Staphylococcus aureus* was the most prevalent bacterial isolate, with 1336 total isolates, including 288 multidrug-resistant (MDR) strains. This aligns with the pathogenesis of *S. aureus*, particularly its methicillin-resistant strain (MRSA), which is a key pathogen in both hospital- and community-acquired infections [11]. *S. aureus* is known for its ability to evade the host’s immune response, colonize wounds, and produce toxins that exacerbate tissue damage. In diabetic patients, the immune system is often compromised due to hyperglycemia, which impairs the ability of white blood cells to fight infection [38]. The high presence of MRSA further complicates treatment due to its resistance to methicillin and other beta-lactam antibiotics, limiting therapeutic options.

The prevalence of *Pseudomonas* spp. (955 isolates) and *Enterobacter* spp. (693 isolates) in this study suggests that Gram-negative bacteria play a significant role in DFI pathogenesis. These organisms are often associated with more severe infections, particularly in patients with compromised immunity and poor wound healing [39]. Pseudomonas, for instance, thrive in moist environments, and their ability to form biofilms on wounds makes it particularly challenging to eradicate. These biofilms protect the bacteria from both the host immune response and antibiotic treatment, contributing to chronic and non-healing ulcers [40].

The high resistance rate in *S. aureus* underscores the importance of local antibiograms and continuous surveillance. MRSA shows resistance despite already being resistant to β-lactams, possibly indicating added resistance to other classes. MSSA resistance suggests the emergence of secondary resistance mechanisms even in methicillin-sensitive strains. The high resistance profile shown by *Pseudomonas* spp. raises concern for treatment options and highlights the need to revise empirical therapy protocols for wound infections, diabetic foot ulcers, and diabetic foot infections or hospital-acquired infections, where *Pseudomonas* is prevalent.

Multidrug resistance (MDR), defined as resistance to two or more antibiotic groups, was particularly prevalent in *Acinetobacter* with 274 isolates. The highest MDR counts among all organisms surveyed, shown by *Acinetobacter* spp., attests to its superbug status due to its resilience in harsh environments and biofilm-forming ability, making it a high-priority target for infection control, thus limiting treatment options [40]. The significantly high levels of MDR displayed by *Staphylococcus aureus*, CoNS, *Streptococcus* spp., and *Enterococcus* spp. are also concerning. These Gram-positive organisms are common in wound infections, bloodstream infections, and surgical sites [39]. Coagulase-negative staphylococci (CoNS), once considered low-virulence, showed high MDR, suggesting a shift in clinical significance especially in prosthetic and catheter-related infections. *Enterococcus* spp. MDR raises concerns about vancomycin-resistant enterococci (VRE). The moderate levels of MDR shown by *Escherichia coli* and *Morganella morganii*, which are likely contributors to urinary tract infections and wound infections, particularly concerning due to ESBL (extended-spectrum β-lactamase) or AmpC production. So, empiric treatment with fluoroquinolones or cephalosporins may be ineffective without culture confirmation. The surprisingly moderate MDR levels showed by *Pseudomonas* spp. in the study population despite its intrinsic resistance may reflect local variability or patient population characteristics of the studies included in the review. Though *Proteus, Klebsiella*, *Enterobacter* spp. exhibited lower levels of MDR, they are still concerning, as these organisms can be reservoirs for plasmid-mediated resistance genes and may evolve into extended-spectrum or carbapenem-resistant strains under selective pressure.

The comparison between frequency of resistance to one and two or more antibiotic groups reveals that mono-resistance may serve as an early warning but does not always correlate with MDR risk. Some pathogens develop MDR silently, while others (e.g., *S. aureus*) show predictable progression of resistance. High-risk organisms for progression to MDR include *Staphylococcus aureus*, CoNS, *Pseudomonas* and *E. coli*, indicating adaptive resistance evolution. Silent MDR threats include *Corynebacterium* spp. and *Enterococcus* spp., indicating their ability to develop resistance rapidly or lack of early detection, while Acinetobacter spp. remains as a critical MDR reservoir. These insights imply that early detection and tailored therapy are essential and that reliance on empiric antibiotics without culture guidance risks under-treating MDR strains. In addition, infection control measures must focus on both highly virulent MDR organisms (e.g., *S. aureus*) and stealth MDR carriers like *Corynebacterium* and *Enterococcus*.

Diabetic foot infections (DFIs) are a serious complication of diabetes, often leading to prolonged hospitalizations and a high risk of limb amputation [28]. Patients with diabetes frequently have comorbid conditions such as diabetic neuropathy, nephropathy (chronic kidney disease, CKD) and hypertension. These comorbidities can worsen peripheral circulation, immune function, and overall health, potentially influencing infection outcomes. Understanding whether nephropathy and hypertension affect the development or severity of antibiotic-resistant infections in DFIs is important for risk stratification and management. Recent studies have explored risk factors for multidrug-resistant organisms (MDROs) in DFIs and characterized the microbiology of DFIs in various populations.

The comorbidity analysis further highlights the complex pathogenesis of DFIs. Neuropathy, present in 18.4% of patients, is a critical factor in the development of DFIs. The pathogenesis of diabetic neuropathy involves prolonged hyperglycemia, which leads to nerve damage through several mechanisms. These include the accumulation of advanced glycation end products (AGEs) and the activation of inflammatory pathways that damage nerve fibers. As a result, patients lose sensation in their feet, making them more susceptible to unnoticed injuries, such as blisters or cuts. These minor injuries can evolve into ulcers, which are slow to heal due to poor circulation and immune dysfunction, making them vulnerable to infection along with 6.5–10 times higher amputation rates compared to non-diabetics [41].

The study also revealed significant rates of previous ulceration (7.3%) and previous amputations (6.5%), which underscores the recurrent nature of DFIs. The pathogenesis of recurrent ulceration often involves a combination of neuropathy, PVD, and impaired immune function. Once an ulcer forms, the underlying neuropathy and poor circulation persist, making it difficult for the wound to heal completely. This sets the stage for recurrent infections, which can lead to further tissue damage and, ultimately, amputation [42].

Cardiovascular complications, such as hypertension (16.5%) and peripheral vascular disease (PVD) (16.0%), play a significant role in the pathogenesis of DFIs by impairing blood flow to the extremities. Chronic hyperglycemia leads to endothelial dysfunction, which reduces nitric oxide production and promotes vasoconstriction. Over time, this results in reduced blood supply to peripheral tissues, impairing oxygen and nutrient delivery to the wound site. This poor circulation not only contributes to delayed wound healing but also fosters a favorable environment for bacterial colonization and growth. Furthermore, PVD leads to tissue ischemia, which contributes to ulcer formation and increases the risk of infection and subsequent amputations [43]. Research data reported on risk factors for MDR infections shows that the impact of hypertension on antibiotic resistance is indirect [44]. Hypertension is a red flag for severe DFU risk and poor healing. Uncontrolled blood pressure can worsen ischemia in the extremities, potentially leading to more severe or chronic wounds that require prolonged antibiotic therapy thereby increasing the risk of antibiotic resistance.

Nephropathy, observed in 10.1% of patients, also contributes to the pathogenesis of DFIs by impairing immune function. Chronic kidney disease (CKD) alters the body’s ability to respond to infection, as the kidneys play a role in maintaining electrolyte balance, detoxifying waste, and regulating immune responses. In CKD patients, immune cells such as neutrophils and macrophages often function less efficiently, reducing their ability to fight infections. Additionally, the buildup of metabolic waste products in patients with CKD can further impair healing processes, making them more prone to severe and chronic infections. Chronic kidney disease, wound chronicity, and prior inpatient care are factors that predispose patients to MRSA (methicillin-resistant Staphylococcus aureus) infection in diabetic foot ulcers (DFUs) [10]. Other recent studies have also suggested that patients with diabetic nephropathy (often requiring more frequent healthcare contact) are more likely to acquire resistant hospital pathogens like MRSA [11]. But other studies have indicated that while CKD can coincide with severe infections, it may not be an independent predictor of resistance once other variables are accounted for [44].

The results of the correlation analysis performed on the data reported in the studies included in this systematic review and meta-analysis shows that patients with nephropathy or hypertension may be at higher risk for antibiotic-resistant Staphylococcal infections. These patients might benefit from alternative empiric antibiotic choices. The correlation could reflect common underlying mechanisms or risk factors. Nephropathy may indirectly contribute to antibiotic resistance development by necessitating frequent antibiotic exposures (e.g., for recurrent infections or hospitalization) and limiting antibiotic choices due to renal impairment. Clinically, in DFI patients with CKD, cautious antibiotic selection is required—many antibiotics need renal dosing adjustments or are contraindicated for fear of nephrotoxicity, potentially narrowing treatment options, which can complicate eradication of resistant organisms. Comorbid nephropathy and hypertension can influence DFI outcomes, even if they are not direct causes of resistance. Patients with diabetic nephropathy often have worse healing and survival outcomes when foot infections occur. As noted, ESRD patients with DFU have very high amputation rates and mortality. One review reported that in diabetics with ESRD, 5-year mortality after a foot ulcer can reach ~50–60% [43], significantly higher than in diabetics without nephropathy. Hypertension, as a marker of cardiovascular disease, may also portend worse healing due to poor tissue perfusion [45].

Finally, smoking plays a critical role in the pathogenesis of diabetes-related complications, particularly DFIs. Current smokers comprised 4.0% of the cohort (*n* = 203), and the impact of smoking on DFIs is profound. Smoking induces vasoconstriction and promotes atherogenesis, further reducing blood flow to the extremities and exacerbating the risk of ulcers. Nicotine and other toxic chemicals in cigarettes increase oxidative stress and inflammation, which damages endothelial cells and accelerates the progression of PVD. Additionally, smoking has been shown to impair insulin sensitivity, leading to poorer glycemic control, which exacerbates all the pathological mechanisms underlying DFIs. Smoking also impairs wound healing by reducing the availability of oxygen and nutrients to tissues, making it more difficult for the body to fight infections and repair damaged tissues. As such, smoking cessation should be considered a vital component of diabetes management to improve patient outcomes and reduce the risk of long-term complications [46].

Limitations: The robustness of our findings is limited by factors like the small number of studies available and gaps in the granularity of the data reported in those studies which in turn increase the risk of bias due to small sample sizes. Subgroup analyses were also performed to explore the influence of different factors on antibiotic resistance rates, including the type of bacterial isolate (e.g., *Staphylococcus aureus* vs. *Pseudomonas* spp.), geographic location of the studies and severity of infections (e.g., ulcers vs. osteomyelitis) which showed erroneous results due to lack of uniform availability of data for each of these criteria across the studies for each bacterial DFI pathogen included in this review. The disproportionate representation by country and gender suggests potential biases or limitations in generalizability. Studies heavily weighted toward a few countries (like China and Malaysia) and predominantly male participants may not fully reflect broader population diversity.

## 5. Conclusions

The landscape of antibiotic resistance in diabetic foot infections presents a complex and evolving challenge for clinicians and healthcare systems. The emergence of multidrug-resistant organisms significantly complicates treatment and worsens clinical outcomes. But despite these challenges, several antibiotics including carbapenems, aminoglycosides, and certain beta-lactam/beta-lactamase inhibitor combinations remain effective against common pathogens. However, the susceptibility patterns to these vary significantly by bacterial species, geographic region, and individual patient factors, underscoring the importance of culture-guided therapy whenever possible.

This article provides a comprehensive overview of antibiotic resistance in DFIs, aiding clinicians in evidence-based decision-making. The results of this systematic review present a thorough summary of the bacterial profiles, antibiotic resistance patterns, and comorbidities associated with diabetic foot infections (DFIs) between 2014 and 2024. High prevalence of MDR necessitates culture-guided therapy, prioritizing sensitivity testing over syndromic or empirical approaches in high-risk populations, especially in chronic wound care and diabetic foot management. Vigilant surveillance, localized antibiograms, and precision antibiotic use are critical to managing this evolving threat. These findings also highlight the need for more geographically and demographically balanced sampling in future research to ensure inclusive and globally relevant conclusions.

The findings from this study underscore the critical role of comorbid conditions—particularly smoking, nephropathy, vascular disease, and previous amputations—in shaping antibiotic resistance profiles among DFI pathogens. Notably, smoking emerged as the most consistent and potent predictor of multidrug resistance across multiple species, with *Proteus* spp. showing the strongest correlations. These associations highlight the importance of risk-stratified antibiotic stewardship. Patients with high-risk comorbidities may require (1) broader-spectrum empiric antibiotic therapy pending culture results, (2) earlier and more frequent culture and sensitivity testing, (3) closer clinical monitoring for signs of treatment failure or relapse, and (4) personalized antimicrobial selection protocols integrated into clinical decision-making systems to mitigate this issue.

Furthermore, the strong association between modifiable lifestyle factors such as smoking and high-level resistance supports public health initiatives aimed at smoking cessation and chronic disease management in diabetic populations. These measures may reduce the burden of multidrug-resistant infections and improve clinical outcomes in this vulnerable patient population.

Comorbidities such as neuropathy, peripheral vascular disease, nephropathy, and hypertension significantly contribute to the development and progression of DFIs. The complex pathogenesis of DFIs involves impaired circulation, immune dysfunction, and delayed wound healing, which creates an environment conducive to recurrent infections and severe complications, including amputation. The data also underscores the detrimental impact of smoking on wound healing and vascular health, further complicating the management of DFIs.

Moving forward, the implementation of multidisciplinary care and antibiotic stewardship programs will be critical to improving patient outcomes. Tailored treatment plans, informed by culture and sensitivity results, will be essential to combat the rise of multidrug-resistant pathogens. Preventive strategies, such as routine foot inspections, early diagnosis, and aggressive management of comorbidities, particularly neuropathy and PVD, will be key in reducing the incidence and severity of DFIs. Additionally, smoking cessation must be integrated into diabetes care to mitigate long-term complications. By recognizing the nuanced impact of comorbidities and adhering to evidence-based practices, it is possible to improve DFI outcomes and mitigate antibiotic resistance in this vulnerable population. Addressing these challenges through holistic, patient-centered approaches can significantly improve the management and treatment outcomes of DFIs. Additionally, preventive measures to reduce the incidence of diabetic foot ulcers and infections remain essential components of any comprehensive strategy to combat resistance in this vulnerable population.

The substantial clinical and economic burden of diabetic foot infections, particularly those caused by resistant organisms, highlights the urgent need for continued attention to this critical area of antimicrobial resistance. Only through coordinated efforts across clinical practice, research, and public health domains can we hope to preserve antibiotic efficacy and improve outcomes for patients with these challenging infections. 

## Figures and Tables

**Figure 1 microorganisms-13-02311-f001:**
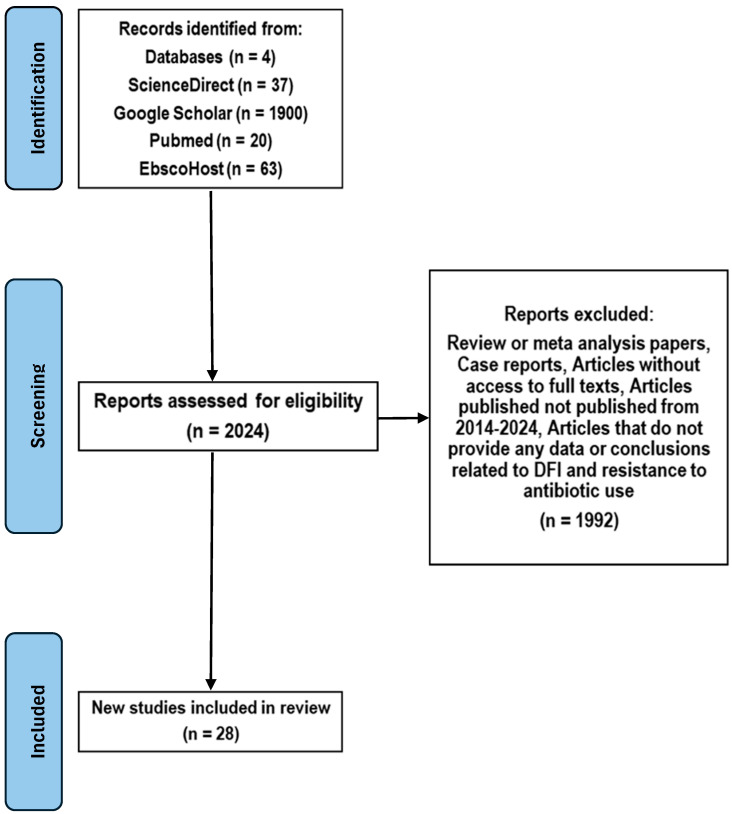
PRISMA Flow Diagram.

**Figure 2 microorganisms-13-02311-f002:**
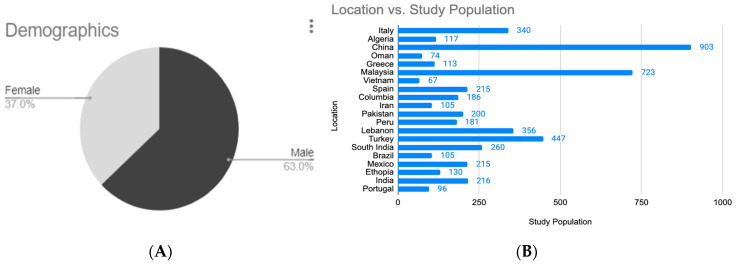
(**A**) Demographic and (**B**) Geographic Distribution of study populations.

**Figure 3 microorganisms-13-02311-f003:**
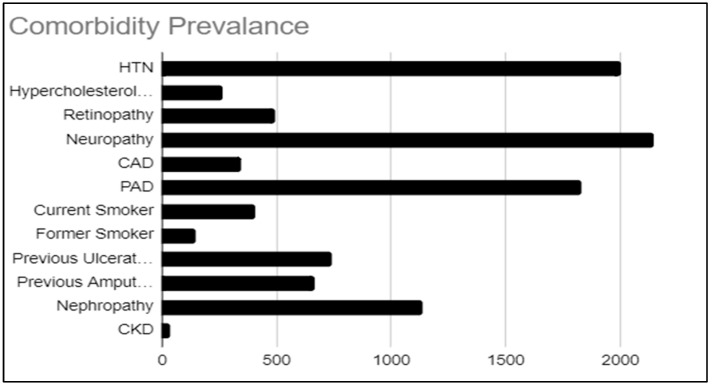
Prevalence of comorbidities in study populations with diabetic foot infections showing antibiotic resistance. HTN, hypertension; CAD, coronary artery disease; PAD, peripheral artery disease; CKD, chronic kidney disease.

**Figure 4 microorganisms-13-02311-f004:**
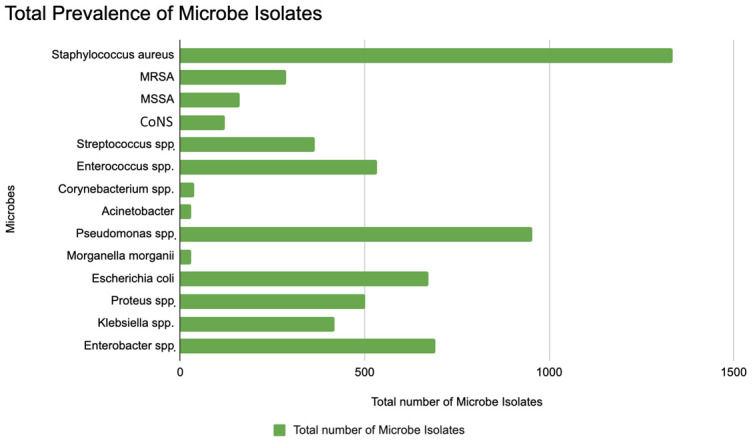
Total Prevalence of Microbial species isolated in study populations with diabetic foot populations showing antibiotic resistance. *Staphylococcus aureus* showed the highest frequency of single antibiotic resistance. MRSA (Methicillin-Resistant *Staphylococcus aureus*); MSSA (Methicillin-Sensitive *S. aureus*); CoNS (Coagulase Negative *Staphyloccus*).

**Figure 5 microorganisms-13-02311-f005:**
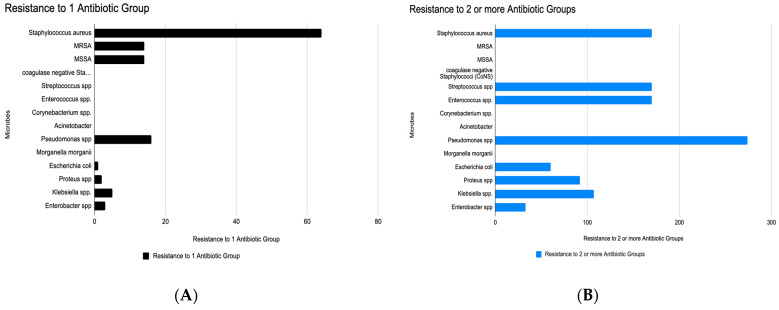
Total Prevalence of (**A**) Single and (**B**) Multi-drug resistance in the various Microbial species isolated in study populations with diabetic foot populations.

**Figure 6 microorganisms-13-02311-f006:**
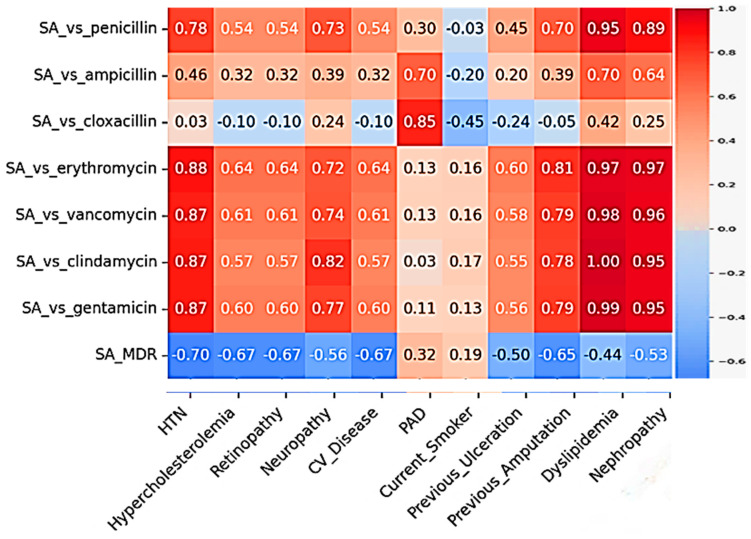
Heatmap showing Pearson correlation coefficients between *Staphylococcus aureus* antibiotic resistance patterns and different clinical comorbidities.

**Table 1 microorganisms-13-02311-t001:** Summary of 28 references reporting the pathogens and their resistance related to diabetic infection.

Search Source	Yield	Articles Included	References
ScienceDirect	37	7	[1,2,3,4,5,6,7]
Pubmed	20	9	[8,13,14,15,16,17,18,19,20,21]
EbscoHost	63	7	[22,23,24,25,26,27,28]
Google Scholar	1900	5	[29,30,31,32,33]

**Table 2 microorganisms-13-02311-t002:** Combined Interpretation: Monoresistance vs. Multidrug Resistance.

Microbe	High in Monoresistance	High in MDR	Key Observations & Clinical Implications
*Staphylococcus aureus*	Very high (~75)	High (~190)	Rapid transition from monoresistance to MDR. Empirical treatment options like β-lactams may fail unless susceptibility is confirmed.
MRSA	Moderate (~15)	Low	Shows low MDR count here, possibly underreported or isolated to β-lactam resistance. Still clinically concerning due to treatment limitations.
MSSA	Moderate (~15)	Low	Growing evidence that even methicillin-sensitive strains are developing additional resistances.
*Coagulase-negative Staphylococci*	Present	High (~160)	Often dismissed as contaminants, but high MDR burden signals emerging pathogenic roles, especially in device-related infections.
*Streptococcus* spp.	Present	High (~160)	High resistance burdens in both categories signal reduced efficacy of penicillin-class drugs in wound and soft tissue infections.
*Enterococcus* spp.	Low	High (~160)	Not prominent in monoresistance but highly represented in MDR—concerning for vancomycin resistance and nosocomial infections.
*Corynebacterium* spp.	Low	Very high (~280)	A striking MDR surge despite low monoresistance. Often under-recognized, but data suggest it may be a significant reservoir of resistance genes.
*Acinetobacter* spp.	Moderate (~25)	Highest (~280)	Dominates MDR list despite modest monoresistance. Known for rapid resistance development and survival in healthcare settings.
*Pseudomonas* spp.	Moderate (~35)	Moderate (~80)	Common culprit in chronic and biofilm-associated infections. Resistance mechanisms include efflux pumps and porin mutations.
*Morganella morganii*	Low	Moderate (~90)	Rarely spotlighted, but its MDR profile is growing, suggesting need for surveillance in polymicrobial infections.
*Escherichia coli*	Low	Moderate (~110)	Resistance likely due to ESBL or AmpC production. Empirical fluoroquinolone or cephalosporin use may be ineffective.
*Proteus, Klebsiella*, *Enterobacter* spp.	Low to moderate	Low to moderate	These organisms can serve as reservoirs of plasmid-mediated resistance, with potential to evolve into extended-spectrum or carbapenem-resistant forms.

**Table 3 microorganisms-13-02311-t003:** Analysis of correlation between types of comorbidities found in staphylococcal infections of the diabetic foot and types of antibiotic resistance.

Comorbidity	Antibiotic	Pearson’s r	*p* Value	Significance
Nephropathy	Penicillin	0.956	0.0051	Significant correlation
	Erythromycin	0.952	0.0047	Significant correlation
	Clindamycin	0.93	0.0032	Significant correlation
Hypertension	Penicillin	0.90	0.0010	Significant correlation
	Erythromycin	0.90	0.0037	Significant correlation
	Clindamycin	0.88	0.0045	Significant correlation
Dyslipidemia	Penicillin	0.95	0.0136	Significant correlation
	Erythromycin	0.97	0.0051	Significant correlation
	Clindamycin	0.99	0.0002	Significant correlation

Significant correlations (|r| > 0.4, *p* < 0.005).

**Table 4 microorganisms-13-02311-t004:** Summary Table: Correlation Between Comorbidities and Antibiotic Resistance in DFI Pathogens.

Pathogen	Comorbidity	Associated Antibiotic Resistance	Strength of Correlation
*S. aureus*	Dyslipidemia	Clindamycin, Gentamicin, Vancomycin, Erythromycin, Penicillin	Very Strong
	Hypertension	Erythromycin	Strong
*Pseudomonas* spp.	Neuropathy	Cephalosporins, Tetracycline, Gentamicin, Quinolones, Carbapenems	Very Strong
	Smoking	Erythromycin, Piperacillin-tazobactam, Imipenem	Strong
	PAD	Carbapenems, Gentamicin, Piperacillin-tazobactam	Moderate
	Nephropathy	Carbapenems, Gentamicin, Piperacillin-tazobactam	Moderate
	Hypertension	Carbapenems, Gentamicin, Erythromycin	Moderate
*Enterococcus* spp.	Smoking	Vancomycin, Tetracycline, Gentamicin	Strong
	Hypertension	Vancomycin, Tetracycline, Gentamicin	Strong
	Nephropathy	Vancomycin, Tetracycline, Gentamicin	Moderate
	PAD	Vancomycin, Tetracycline, Gentamicin	Moderate
*E. coli*	Smoking	Piperacillin-tazobactam, Imipenem, Cephalosporins, Quinolones, Penicillin, Gentamicin	Strong
	Nephropathy	Aminoglycosides, Carbapenems, Tetracycline, Gentamicin	Moderate
	Amputation	Aminoglycosides, Carbapenems, Tetracycline, Gentamicin	Moderate
	Hypertension	Penicillin, Tetracycline, Gentamicin	Moderate
*Proteus* spp.	Current Smoking	Erythromycin, Amoxicillin/clavulanate, Piperacillin-tazobactam, Imipenem, Cephalosporins, Quinolones, Penicillin, Gentamicin	Very Strong
	Previous Amputation	Gentamicin, Penicillin, Erythromycin, multiple others	Very Strong
	Nephropathy	Aminoglycosides	Strong
	Former Smoker	Tetracycline	Strong

## Data Availability

No new data were created or analyzed in this study. Data sharing is not applicable to this article.

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
