# Peer review of "Multifaceted Antibiotic Resistance in Diabetic Foot Infections: A Systematic Review"

_microorganisms, 2025, doi:10.3390/microorganisms13102311_

Round 1
Reviewer 1 Report
Comments and Suggestions for Authors
Dear authors, I had the pleasure of reading your manuscript entitled "Multifaceted Antibiotic Resistance in Diabetic Foot Infections: 2 A Systematic Review.” After reading your manuscript, I have some comments.
Figure 1 has not been included in the text.
Line 179 Figure 1 should be Figure 2A.
Please include the abbreviations for Figure 3 in the figure captions.
Lines 197 and 269 diabetic foot infections have been previously abbreviated.
All bacteria names must be in italics. Please check this throughout the text.
Figure 4 and Figure 5 should be Morganella morganii and not Morganella morangii
In Figure 4, the proportion of Acinetobacter and Morganella morganii appears to be less than 100 isolates; however, in Figure 5b, the proportion of Acinetobacter isolates appeared to be almost 300 and that of Morganella morganii was almost 100.
The authors considered bacteria to be multidrug-resistant when they were resistant to two or more antibiotics. According to scientific literature, they are considered multidrug-resistant when they are resistant to three or more chemical groups of antibiotics.
Table 2 is unclear. Why does it have question marks “High in Monoresistance?” “High in MDR?” Why are the numbers indicated approximate?
Table 3 is not cited in the text.
The figure on page 8 has no caption.
The heatmaps are not clear; please improve the resolution.
The conclusion is too long, and some paragraphs can be found in the Discussion section.
The most critical point of this article is the discrepancy in the data on the number of isolates in the figures. Please check the data obtained from the manuscripts very carefully.
Author Response
- Figure 1 has not been included in the text.
Response: Figure 1 is a part of the selection process, since PRISMA is standard procedures
- Line 179 Figure 1 should be Figure 2A.
Response: This has been fixed as suggested
- Please include the abbreviations for Figure 3 in the figure captions.
- Response: This has been fixed as suggested
- Lines 197 and 269 diabetic foot infections have been previously abbreviated.
Response: This has been fixed as suggested
- All bacteria names must be in italics. Please check this throughout the text.
Response: This has been fixed as suggested
- Figure 4 and Figure 5 should be Morganella morganii and not Morganella morangii
Response: Thank you for your feedback. This has been fixed as suggested
- In Figure 4, the proportion of Acinetobacter and Morganella morganii appears to be less than 100 isolates; however, in Figure 5b, the proportion of Acinetobacter isolates appeared to be almost 300 and that of Morganella morganii was almost 100.
Response: This has been fixed as suggested
- The authors considered bacteria to be multidrug-resistant when they were resistant to two or more antibiotics. According to scientific literature, they are considered multidrug-resistant when they are resistant to three or more chemical groups of antibiotics.
Response: Thank you for your feedback. The following literature has defined MDR being resistant to two or more bacteria
- Giske CG, Monnet DL, Cars O, Carmeli Y; ReAct-Action on Antibiotic Resistance. Clinical and economic impact of common multidrug-resistant gram-negative bacilli. Antimicrob Agents Chemother. 2008 Mar;52(3):813-21. doi: 10.1128/AAC.01169-07. Epub 2007 Dec 10. PMID: 18070961; PMCID: PMC2258516.
- Gabriel G. Perron, Graham Bell, Sylvain Quessy, Parallel evolution of multidrug-resistance in Salmonella entericaisolated from swine, FEMS Microbiology Letters, Volume 281, Issue 1, April 2008, Pages 17–22, https://doi.org/10.1111/j.1574-6968.2007.01045.x
- Mouiche, M.M.M., Moffo, F., Akoachere, JF.T.K. et al.Antimicrobial resistance from a one health perspective in Cameroon: a systematic review and meta-analysis. BMC Public Health 19, 1135 (2019). https://doi.org/10.1186/s12889-019-7450-5
- Table 2 is unclear. Why does it have question marks “High in Monoresistance?” “High in MDR?” Why are the numbers indicated approximate?
Response: The question marks have been removed. The numbers included in the table are approximate because qualitative terms such as “moderate,” “notable,” and “low” are used to describe resistance levels for various organisms, these descriptors must be quantified approximately to enable comparison across species. Additionally, for several organisms, resistance levels are given as ranges rather than discrete counts. When summarizing the data in a table, midpoint approximations were used to reflect the range.
- Table 3 is not cited in the text.
Response: Corrected. Cited in text now in results section
- The figure on page 8 has no caption.
Response: A caption has now been added
- The heatmaps are not clear; please improve the resolution.
Response: A clearer version has now been added. Additionally, please refer to the values from the supplementary table, and the colors of the blocks in the heatmap represent the degree of correlation between comorbidities and resistance variable for each pathogen based on the R value.
- The conclusion is too long, and some paragraphs can be found in the Discussion section.
Response: We have revised and shortened the conclusion section to eliminate redundancy with the discussion section.
- The most critical point of this article is the discrepancy in the data on the number of isolates in the figures. Please check the data obtained from the manuscripts very carefully.
Response: Corrected
Reviewer 2 Report
Comments and Suggestions for Authors
There are some minor revisions regarding the chapters Results and Discussion:
Results:
- Edit the name of Morganella morganni in Fig.4 , Fig 5
- Since the whole description for the correlation between the microbial agents and their resistance to different classes of antibiotics is in the supplementary files, can you include in the main text a concise overview of the most significant resistant antibiotic classes associated with the discussed bacterial agents. For example, highlight the top three major antimicrobial classes showing resistance in Staphylococcus aureus, Enterococcus, Pseudomonas, etc. It would be beneficial to present this information either in brackets next to the cited bacterial agents or in a table format. This will help readers easily identify the most problematic antimicrobial drugs for these bacterial agents and where the highest resistance occurs.
- Regarding the resistance to two or more antibiotic groups, which analyze the burden of multidrug resistance (MDR) in Line 223, Line 302, the definition for multidrug resistance or multiresistance is antimicrobial resistance shown by a species of microorganism to at least one antimicrobial drug in three or more antimicrobial categorie, not two.
- Figures Correlation heat maps – The visualization does not display well in this format.
Discussion:
Add a few sentences about which classes of antimicrobials display the biggest challenges and lead to complicated therapy for patients with diabetes.
Author Response
Results:
- Edit the name of Morganella morganni in Fig.4 , Fig 5
Response: Fixed
- Since the whole description for the correlation between the microbial agents and their resistance to different classes of antibiotics is in the supplementary files, can you include in the main text a concise overview of the most significant resistant antibiotic classes associated with the discussed bacterial agents. For example, highlight the top three major antimicrobial classes showing resistance in Staphylococcus aureus, Enterococcus, Pseudomonas, etc. It would be beneficial to present this information either in brackets next to the cited bacterial agents or in a table format. This will help readers easily identify the most problematic antimicrobial drugs for these bacterial agents and where the highest resistance occurs.
Response: We have added a concise overview of the most significant resistant antibiotic classes associated with the discussed bacterial agents, in the main text.
We have not performed analysis regression between the pathogen and resistance type, so we are unable to identify the strongest, most significant correlation values for resistance to antibiotic classes associated with each type of bacterial agent. The focus of our paper is to identify and predict which types of pathogen occurrence and comorbidities could be strongly correlated to significant resistance against major antibiotics in DFI patients rather than the type of antibiotic resistance.
- Regarding the resistance to two or more antibiotic groups, which analyze the burden of multidrug resistance (MDR) in Line 223, Line 302, the definition for multidrug resistance or multiresistance is antimicrobial resistance shown by a species of microorganism to at least one antimicrobial drug in three or more antimicrobial categories, not two.
Response: Thank you for your feedback. The following literature has defined MDR being resistant to two or more bacteria
- Giske CG, Monnet DL, Cars O, Carmeli Y; ReAct-Action on Antibiotic Resistance. Clinical and economic impact of common multidrug-resistant gram-negative bacilli. Antimicrob Agents Chemother. 2008 Mar;52(3):813-21. doi: 10.1128/AAC.01169-07. Epub 2007 Dec 10. PMID: 18070961; PMCID: PMC2258516.
- Gabriel G. Perron, Graham Bell, Sylvain Quessy, Parallel evolution of multidrug-resistance in Salmonella entericaisolated from swine, FEMS Microbiology Letters, Volume 281, Issue 1, April 2008, Pages 17–22, https://doi.org/10.1111/j.1574-6968.2007.01045.x
- Mouiche, M.M.M., Moffo, F., Akoachere, JF.T.K. et al.Antimicrobial resistance from a one health perspective in Cameroon: a systematic review and meta-analysis. BMC Public Health 19, 1135 (2019). https://doi.org/10.1186/s12889-019-7450-5
- Figures Correlation heat maps – The visualization does not display well in this format.
Response: A clearer version has now been added. Additionally, please refer to the values from the supplementary table, and the colors of the blocks in the heatmap represent the degree of correlation between comorbidities and resistance variable for each pathogen based on the R value.
Discussion:
- Add a few sentences about which classes of antimicrobials display the biggest challenges and lead to complicated therapy for patients with diabetes.
Response: We thank the reviewer for this valuable suggestion. We have added several sentences addressing antimicrobial classes that present the greatest therapeutic challenges in diabetic patients. Furthermore, we have incorporated analysis of drug resistance patterns associated with diabetes comorbidities for key problematic pathogens including Staphylococcus aureus, Enterococcus species, Pseudomonas aeruginosa, Escherichia coli, and Proteus species. The correlation between antimicrobial resistance patterns and comorbidity status is demonstrated through our heat map analysis, which illustrates how resistance to these critical antimicrobial classes significantly complicates therapeutic decision-making in diabetic patients.
These additions directly address the intersection of antimicrobial class limitations and bacterial resistance patterns that create the most complex therapeutic scenarios for diabetic patients, which aligns with the primary scope and objectives of our paper.
Reviewer 3 Report
Comments and Suggestions for Authors
Introduction
- The aim of the review is mentioned in the Introduction (Lines 72-78), as well as in Methods (Lines82-84). I suggest mentioning the aim of the review only in the Introduction.
- Please revise Lines 76-78 (“Additionally, it emphasizes the importance of integrating antibiotic stewardship principles into clinical practice to combat resistance and improve therapeutic outcomes in the treatment of DFIs as the lines belong to discussions.
- There are no references used in the Introduction section.
Methods
- Data extraction: please mention the initials of the authors that performed data extraction.
- What tool was used for assessing the risk of Bias? The authors mention that “The risk of bias in individual studies was assessed using the Cochrane risk of bias tool for non-randomized studies. „, but Chrochare recommended different tools for assessing the Risk of Bias. Please name the tool that was used. How many researchers assessed ROB?
- Please clarify if a meta-analysis was performed. The I² is a statistic test used in meta-analysis..
- The authors state that „The quality of the included studies was assessed using the PRISMA (Preferred Reporting Items for Systematic Reviews and Meta-Analyses) guidelines.”, but the PRISMA guidelines are setting reporting standards. The quality of the included studies has to be assessed with the help of other tools. Please revise.
- Please add a PRISMA 2020 checklist in the supplementary materials, if the review adheres to the PRISMA guidelines.
Results:
- The PRISMA flowchart is typically presented in the Results section. The results section starts with a table that is not necessary if the PRISMA flow chart is added to the results section.
- The PRISMA flowchart needs to be revised to present the screening process more clearly. How many articles were excluded after the title screening? How many articles were excluded after abstract screening? How many articles were excluded after full-text screening?
- Figure 2 presents the demographics. It is unclear how many patients are from which country.
- Figure 3 is unclear. The legend cannot be read.
- The results section needs to be revised to increase transparency. References (of the included studies) are missing and the numbers cannot be verified.
E.g. Lines 193-196: “Among the 5096 patients included in the study, the most prevalent comorbidity was neuropathy, affecting 18.4% of the cohort (n = 939). Hypertension (HTN) was the second most common, present in 16.5% of patients (n = 839), followed closely by peripheral vascular disease, which was observed in 16.0% of the population (n = 816).”
- What study? Are the 5096 patients pooled from all included studies?
- 18.4% out of 5096 is 937.66 (938), 16% out of 5096 is 815.36 (rounded up to 815). Please check the numbers
- Figure 4 presents the total prevalence of isolated bacteria. It is unclear from what studies the data was derived. Multiple figures have to be revised (duplicate data, as it presents S. aureus separately and then again MSSA and MRSA, spp. – the dot is missing sometimes, etc.). Please also add legends.
- The bacteria names should it italicized in the entire document
- Some lines (e.g. Lines 121-122) interpret results, hence they belong to the Discussion section
- The results section contains unclear and, sometimes unnamed figures (e.g. Figure S5 is unreadable)
- The manuscript is a systematic review that includes 28 studies that are not cited in the results section.
- The authors assess the ROB, but the results are not mentioned in the manuscript, nor the supplementary materials.
Overall observations: The manuscript presents several methodological flows that have to be addressed. The results section has to be revised, preferably according to the PRISMA 2020 checklist.
Author Response
Introduction
- The aim of the review is mentioned in the Introduction (Lines 72-78), as well as in Methods (Lines82-84). I suggest mentioning the aim of the review only in the Introduction.
Response: We have removed the aim statement from the Methods section (Lines 82–84) and retained it in the Introduction (Lines 72–78).
- Please revise Lines 76-78 (“Additionaly, it emphasizes the importance of integrating antibiotic stewardship principles into clinical practice to combat resistance and improve therapeutic outcomes in the treatment of DFIs as the lines belong to discussions.
Response: That sentence has been revised for clarity - There are no references used in the Introduction section.
Response: References added, as suggested
Methods
- Data extraction: please mention the initials of the authors that performed data extraction.
Response: We have updated the Methods section to include the initials of the authors who performed data extraction.
- What tool was used for assessing the risk of Bias? The authors mention that “The risk of bias in individual studies was assessed using the Cochrane risk of bias tool for non-randomized studies. „, but Chrochare recommended different tools for assessing the Risk of Bias. Please name the tool that was used. How many researchers assessed ROB?
- Response: ROBINS is the tool used for risk of bias analysis. Table added to supplemental section. Methods have been updated as suggested.
- Please clarify if a meta-analysis was performed. The I² is a statistic test used in meta-analysis..
Response: This study was originally planned as a meta-analysis; however, due to insufficient data for a reliable risk of bias assessment, we did not proceed with a full meta-analysis
- The authors state that „The quality of the included studies was assessed using the PRISMA (Preferred Reporting Items for Systematic Reviews and Meta-Analyses) guidelines.”, but the PRISMA guidelines are setting reporting standards. The quality of the included studies has to be assessed with the help of other tools. Please revise.
Response: Quality assessment process description has been added in the methods section and the Cochrane ROBINS tool was used for risk of bias assessment which has been added to the supplement section
- Please add a PRISMA 2020 checklist in the supplementary materials, if the review adheres to the PRISMA guidelines.
Response: The PRISMA 2020 checklist has been added to the supplementary materials
Results:
- The PRISMA flowchart is typically presented in the Results section. The results section starts with a table that is not necessary if the PRISMA flow chart is added to the results section.
The PRISMA flowchart needs to be revised to present the screening process more clearly. How many articles were excluded after the title screening? How many articles were excluded after abstract screening? How many articles were excluded after full-text screening?
Response: We have referred to the standard PRISMA guidelines for reporting a systematic review and the flowchart follows the PRISMA 2020 statement. The PRISMA checklist is provided in the supplementary materials.
- Figure 2 presents the demographics. It is unclear how many patients are from which country.
Response: Figure 2B states the number/size of the study populations from each country or location
- Figure 3 is unclear. The legend cannot be read.
Response: Corrected
- The results section needs to be revised to increase transparency. References (of the included studies) are missing and the numbers cannot be verified.
E.g. Lines 193-196: “Among the 5096 patients included in the study, the most prevalent comorbidity was neuropathy, affecting 18.4% of the cohort (n = 939). Hypertension (HTN) was the second most common, present in 16.5% of patients (n = 839), followed closely by peripheral vascular disease, which was observed in 16.0% of the population (n = 816).”
Response: Results section has been revised
- What study? Are the 5096 patients pooled from all included studies? 18.4% out of 5096 is 937.66 (938), 16% out of 5096 is 815.36 (rounded up to 815). Please check the numbers
Refers to this study. Yes all included studies.
- Figure 4 presents the total prevalence of isolated bacteria. It is unclear from what studies the data was derived. Multiple figures have to be revised (duplicate data, as it presents S. aureus separately and then again MSSA and MRSA, spp. – the dot is missing sometimes, etc.). Please also add legends.
Response: Legends have been added
- The bacteria names should it italicized in the entire document
Response: Done
- Some lines (e.g. Lines 121-122) interpret results, hence they belong to the Discussion section
The results section contains unclear and, sometimes unnamed figures (e.g. Figure S5 is unreadable)
Response: A clearer version has now been added. Additionally, please refer to the values from the supplementary table, and the colors of the blocks in the heatmap represent the degree of correlation between comorbidities and resistance variable for each pathogen based on the R value.
- The manuscript is a systematic review that includes 28 studies that are not cited in the results section.
Response: Done
- The authors assess the ROB, but the results are not mentioned in the manuscript, nor the supplementary materials.
Response: ROBINS table has been added in the supplement section
Overall observations: The manuscript presents several methodological flows that have to be addressed. The results section has to be revised, preferably according to the PRISMA 2020 checklist.
Reviewer 4 Report
Comments and Suggestions for Authors
Regarding to manuscript microorganisms-3659207, entitled “Multifaceted Antibiotic Resistance in Diabetic Foot Infections: A Systematic Review”, authors summarized the data from 28 references that reported the pathogens and their resistamce related to diabetic infection.
This review manuscript shall be revised completely as the following suggestion.
- Authors shall remove the methods to the last section and define the multiple drug resistance (MDR)
- Authors shall rename the results on each topics;
- epidemiological analysis of
- pathogens related to comorbidity especially to cause death
- pathogens related to antibiotic resistance
- please italicize the scientific name through all text
- please rename the statistic data to 0.001 no more less than that throughout the
text.
- Please revise Figure S5 to a Table that only showed significantly differences.
Such as use penicillin instead of SA vs penicillin.
- Please present a Conclusion to present the finding different from previous reports.
.
- Authors shall revise the supplementary data by disease pattern and add the
References.
- Authors shall have a supplementary table to summarize all the data based on references disease patterns, number, and so on.
Author Response
This review manuscript shall be revised completely as the following suggestion.
- Authors shall remove the methods to the last section and define the multiple drug resistance (MDR)
Response: Done
- Authors shall rename the results on each topics
Response: Done
- epidemiological analysis of- pathogens related to comorbidity especially to cause death - pathogens related to antibiotic resistance
- please italicize the scientific name through all text
- please rename the statistic data to 0.001 no more less than that throughout the text.
Response: Done
- Please revise Figure S5 to a Table that only showed significantly differences.
- Response: Thank you for your valuable suggestions. We are happy to provide the table upon request but we strongly feel that the figure with the color coding is a visual indicator of the degree of correlation which makes a stronger impact for the same data set.
Such as use penicillin instead of SA vs penicillin.
Response: The tables are generated after analysis by a free software. We do not have the financial means to use a paid software which allows for better customization so we cannot change the nomenclature
- Please present a Conclusion to present the finding different from previous reports.
Response: Done
- Authors shall revise the supplementary data by disease pattern and add the
Response: We are unable to provide this as we are not clear about this comment. We have added a summary table on the degree of correlation of disease and resistance of each DFI pathogen.
References.
- Authors shall have a supplementary table to summarize all the data based on references disease patterns, number, and so on.
Response: We are unable to provide this as we are not clear about this comment. We have added a summary table on the degree of correlation of disease and resistance of each DFI pathogen.
Round 2
Reviewer 1 Report
Comments and Suggestions for Authors
the manuscript was improved
Author Response
Thank you for your encouragement and constructive comments/ suggestions to improve the manuscript. We highly appreciate your motivating comments.
Sincerely,
Weiqi Li, Oren Sadeh, Jina Chakraborty, Emily Yang, Paramita Basu and Priyank Kumar
Reviewer 3 Report
Comments and Suggestions for Authors
The revised version of “Multifaceted Antibiotic Resistance in Diabetic Foot Infections: A Systematic Review” still lacks transparency from a methodological perspective. While the authors have addressed some of the previous concerns, several important issues remain:
- Introduction – The section now contains two references; however, many statements are still unsupported by references (starting from line 44).
- Search strategy – The authors state that the search was conducted in four databases, but lines 91–97 (in the Methods section) present results from only three. There are inconsistencies in the reported numbers and misplacement of information across sections.
- Section structure – Misplacements extend into subsections. For example, the Data Extraction section begins with Risk of Bias Assessment, which is inappropriate.
- Bacterial nomenclature – While bacterial names are italicized, they are still not fully consistent with international recommendations (e.g., “Pseudomonas spp.” should be used instead of “Pseudomonas spp.”).
- Meta-analysis – Lines 172–173 refer to a meta-analysis. However, in their reply, the authors state: “This study was originally planned as a meta-analysis; however, due to insufficient data for a reliable risk of bias assessment, we did not proceed with a full meta-analysis.” This inconsistency should be resolved.
- PRISMA flow chart – A flow chart was added, but it is incomplete. Numbers do not add up (4 + 37 + 1900 + 20 + 63 = 2024, not 2020). The flow diagram, along with related descriptions, contains several inconsistencies.
- References – References should be placed before the period, in line with standard citation practices.
- Figure 4 – This figure still presents duplicate data (S. aureus is shown separately, but also again as MSSA and MRSA).
- Abstract – The phrase “a total of 28 studies” is redundant. Rephrasing is recommended.
Many other issues are presents around the manuscript and I do not recommend publishing. While minor observations can be corrected, the discrepencies between the numbers, misplacing of the information on section, methodological flow require extensive revisions.
Author Response
Comments and Suggestions for Authors
The revised version of “Multifaceted Antibiotic Resistance in Diabetic Foot Infections: A Systematic Review” still lacks transparency from a methodological perspective. While the authors have addressed some of the previous concerns, several important issues remain:
- Introduction – The section now contains two references; however, many statements are still unsupported by references (starting from line 44).
— References added
2. Search strategy – The authors state that the search was conducted in four databases, but lines 91–97 (in the Methods section) present results from only three. There are inconsistencies in the reported numbers and misplacement of information across sections.
-Corrected
3. Section structure – Misplacements extend into subsections. For example, the Data Extraction section begins with Risk of Bias Assessment, which is inappropriate.
-order of placement of subsections have been changed. Now the data extraction section is separated followed by a separate risk of bias analysis section on.
4. Bacterial nomenclature – While bacterial names are italicized, they are still not fully consistent with international recommendations (e.g., “Pseudomonas spp.” should be used instead of “Pseudomonas spp.”).
-Corrected
5. Meta-analysis – Lines 172–173 refer to a meta-analysis. However, in their reply, the authors state: “This study was originally planned as a meta-analysis; however, due to insufficient data for a reliable risk of bias assessment, we did not proceed with a full meta-analysis.” This inconsistency should be resolved.
-The reply referred to explanations about the earlier version
6. PRISMA flow chart – A flow chart was added, but it is incomplete. Numbers do not add up (4 + 37 + 1900 + 20 + 63 = 2024, not 2020). The flow diagram, along with related descriptions, contains several inconsistencies.
-Corrected to 2024 but we did not find any other inconsistency in the PRISMA diagram
7. References – References should be placed before the period, in line with standard citation practices.
-Corrected
8. Figure 4 – This figure still presents duplicate data (S. aureus is shown separately, but also again as MSSA and MRSA).
-This data cannot be differentiated as the included studies report pathogens in the patients as S. aureus separately and then again MSSA and MRSA so we have represented them accordingly to avoid misrepresentation of the data.
9. Abstract – The phrase “a total of 28 studies” is redundant. Rephrasing is recommended.
-We strongly feel this is relevant in the abstract as it is important to state the total number of studies from which data is extracted to provide context
We respectfully acknowledge the reviewer’s concerns and thank them for their detailed feedback. However, we strongly feel that the revised manuscript has addressed the major methodological and structural issues raised in the first round of review. We have made substantial improvements, including:
-
Correcting the section structure for clarity and logical flow,
-
Ensuring consistency in bacterial nomenclature,
-
Clarifying the search strategy and database coverage,
-
Resolving numerical inconsistencies in the PRISMA flowchart,
-
Providing justifications for data representation in Figure 4, and
-
Enhancing transparency in both the abstract and methods.
We believe that the current version of the manuscript demonstrates methodological rigor, clarity of presentation, and clinical relevance in the context of antibiotic resistance in diabetic foot infections. We respectfully submit that the revised manuscript merits reconsideration and should be accepted for publication following a second round of review, during which we remain open to further refinements based on editorial or reviewer input.